# Differentiable Cloth Simulation for Inverse Problems

**Junbang Liang**          **Ming C. Lin**
University of Maryland, College Park

**Vladlen Koltun**
Intel Labs

## Abstract

We propose a differentiable cloth simulator that can be embedded as a layer in deep neural networks. This approach provides an effective, robust framework for modeling cloth dynamics, self-collisions, and contacts. Due to the high dimensionality of the dynamical system in modeling cloth, traditional gradient computation for collision response can become impractical. To address this problem, we propose to compute the gradient directly using QR decomposition of a much smaller matrix. Experimental results indicate that our method can speed up backpropagation by two orders of magnitude. We demonstrate the presented approach on a number of inverse problems, including parameter estimation and motion control for cloth.

## 1   Introduction

Differentiable physics simulation is a powerful family of techniques that applies gradient-based methods to learning and control of physical systems [7; 8; 29; 13; 25]. It can enable optimization for control, and can also be integrated into neural network frameworks for performing complex tasks. Our work focuses on cloth simulation, which relates to applications in robotics, computer vision, and computer graphics [6; 19; 3; 31; 23; 16; 11]. Our goal is to enable differentiable cloth simulation, which can provide a unified approach to a variety of inverse problems that involve cloth.

Differentiable cloth simulation is challenging due to a number of factors, which include the high dimensionality of cloth (as compared for example to rigid bodies [7]) and the need to handle contacts and collision. For example, a simple 16×16 grid-based cloth mesh has 289 vertices, 867 variables, and 512 faces when triangulated. Typical resolutions for garments would be at least many thousands, if not millions, of vertices and faces. Previous work that tackled differentiable simulation with collisions set up a static linear solver to account for all constraints [7]. In our simple example with cloth, the number of pairwise constraints would be at least $289 \times 512 = 140K$ for vertex-face collisions alone, which renders existing methods impractical even for this simple system. Even if a dynamic solver is applied upon collision, solving a dense linear system with such high dimensionality makes the gradient computation infeasible.

In this paper, we propose a differentiable cloth simulation algorithm that overcomes the above difficulties. First, we use dynamic collision detection since the actual collision pairs are very sparse. The collision response is solved by quadratic optimization, for which we can use implicit differentiation to compute the gradient. We directly solve the equations derived from implicit differentiation by using the QR decomposition of the constraint matrix, which is much smaller than the original linear system and is often of low rank. This approach reduces the gradient computation to a linear system of a small upper triangular matrix (the R component of the decomposition), which enables fast simulation and backpropagation.

Our experiments indicate that the presented method makes differentiable cloth simulation practical. Using our method, the largest size of the linear system is 10x-20x smaller than the original solver in the backpropagation of the collision response, and the solver is 60x-130x faster. We demonstrate the potential of differentiable cloth simulation in a number of application scenarios, such as physical parameter estimation and motion control of cloth. With only a few samples, the differentiable

simulator can optimize its input variables to fit the data, thereby inferring physical parameters from observations and reaching desired control goals.

## 2 Related Work

**Differentiable physics.** With recent advances in deep learning, there has been increasing interest in differentiable physics simulation, which can be combined with other learning methods to provide physically consistent predictions. Belbute-Peres *et al*. [7] and Degrave *et al*. [8] proposed rigid body simulators using a static formulation of the linear complementarity problem (LCP) [5; 4]. Toussaint *et al*. [29] developed a robot reasoning system that can achieve user-defined tasks and is based on differentiable primitives. Hu *et al*. [13] implemented a differentiable simulator for soft robots based on the Material Point Method (MPM). They store the object data at every simulation step so that the gradient can be computed out of the box. Schenck and Fox [25] embedded particle-based fluid dynamics into convolutional neural networks, with precomputed signed distance functions for collision handling. They solved or avoided collisions by assuming special object shapes, transferring to an Eulerian grid, or solving the corresponding collision constraint equation.

None of these methods can be applied to cloth simulation. First, cloth is a 2D surface in a 3D world; thus methods that use an Eulerian grid to compute material density, such as MPM [13], are not applicable. Second, the collision constraints in cloth simulation are more dynamic and complex given the high number of degrees of freedom; thus constructing a static dense LCP for the entire system [7; 8] or constructing the overall state transition graph [29] is inefficient and usually impossible for cloth of common resolution, since contact can happen for every edge-edge or vertex-face pair. Lastly, the shape of cloth changes constantly so self-collision cannot be handled by precomputed signed distance functions [25].

In contrast, our method uses dynamic collision detection and computes the gradients of the collision response by performing implicit differentiation on the quadratic optimization used for computing the response. We utilize the low dimensionality and rank of the constraint matrix in the quadratic optimization and minimize the computation needed for the gradient propagation by giving an explicit solution to the linear system using QR decomposition of the constraint matrix.

**Deep learning and physics.** Supervised deep networks have been used to approximate physical dynamics. Mrowca *et al*. [21] and Li *et al*. [17] learned interaction networks to model particle systems. Ingraham *et al*. [14] trained a model to predict protein structures from sequences using a learnable simulator; the simulator predicts the deformation energy as an approximation to the physical process. Deep networks have also been used to support the simulation of fluid dynamics [28; 15; 20]. Our method differs from many works that use deep networks to approximate physical systems in that we backpropagate through the true physical simulation. Thus our method conforms to physical law regardless of the scale of the problem. It can also naturally accept physical parameters as input, which enables learning from data.

**Deep learning and cloth.** Coupling cloth simulation with deep learning has become a popular way to solve problems such as detail refinement, garment retargeting, and material estimation. Yang *et al*. [31] proposed a recurrent model to estimate physical cloth parameters from video. Lähner *et al*. [16] trained a GAN to generate wrinkles on a coarse garment mesh which can then be automatically registered to a human body using PCA. Gundogdu *et al*. [11] trained a graph convolutional framework to generate drapes and wrinkles given a roughly registered mesh. Santesteban *et al*. [24] developed an end-to-end retargeting network using a parametric human body model with displacements to represent the cloth.

These applications may benefit from our method. For garment retargeting problems, the relationship between body pose and vertex displacement is made explicit via the computed gradient, which can then be applied in network regularization for better performance. For parameter estimation, the differentiable simulation provides an optimization-based solution rather than a learning-based one. Instead of learning statistics from a large amount of data, we can directly apply gradient-based optimization via the simulator, which does not require any training data.

# 3 Differentiable Cloth Simulation

In this section, we introduce the main algorithms for the gradient computation. In general, we follow the computation flow of the common approach to cloth simulation: discretization using the finite element method [9], integration using implicit Euler [2], and collision response on impact zones [22; 12]. We use implicit differentiation in the linear solve and the optimization in order to compute the gradient with respect to the input parameters. The discontinuity introduced by the collision response is negligible because the discontinuous states constitute a zero-measure set. During the backpropagation in the optimization, the gradient values can be directly computed after QR decomposition of the constraint matrix.

## 3.1 Overview

We begin by defining the problem formally and providing common notation. A triangular mesh $\mathcal{M} = \{\mathcal{V}, \mathcal{E}, \mathcal{F}\}$ consists of sets of vertex states, edges, and faces, where the state of the vertices includes both position $\mathbf{x}$ and velocity $\mathbf{v}$. Given a cloth mesh $\mathcal{M}_t$ together with obstacle meshes $\mathcal{M}_t^{obs}$ at step $t$, a cloth simulator can compute the mesh state $\mathcal{M}_{t+1}$ at the next step $t+1$ based on the computed internal and external forces and the collision response. A simple simulation pipeline is shown in Algorithm 1, where $\mathbf{M}$ is the mass matrix, $\mathbf{f}$ is the force, and $\mathbf{a}$ is the acceleration. For more detailed description of cloth simulation, please refer to Appendix B. All gradients except the linear solve (Line 4 in Algorithm 1) and the collision response (Line 7) can be computed using automatic differentiation in PyTorch [26].

---

**Algorithm 1** Cloth simulation

1: $\mathbf{v}_0 \leftarrow \mathbf{0}$
2: **for** $t = 1$ **to** $n$ **do**
3:      $\mathbf{M}, \mathbf{f} \leftarrow$ compute_forces($\mathbf{x}, \mathbf{v}$)
4:      $\mathbf{a}_t \leftarrow \mathbf{M}^{-1}\mathbf{f}$
5:      $\mathbf{v}_t \leftarrow \mathbf{v}_{t-1} + \mathbf{a}_t \Delta t$
6:      $\mathbf{x}_t \leftarrow \mathbf{x}_{t-1} + \mathbf{v}_t \Delta t$
7:      $\mathbf{x}_t \leftarrow \mathbf{x}_t +$ collision_response($\mathbf{x}_t, \mathbf{v}_t, \mathbf{x}_t^{obs}, \mathbf{v}_t^{obs}$)
8:      $\mathbf{v}_t \leftarrow (\mathbf{x}_t - \mathbf{x}_{t-1})/\Delta t$
9: **end for**

---

## 3.2 Derivatives of the Physics Solve

In modern simulation algorithms, implicit Euler is often used for stable integration results. Thus the mass matrix $\mathbf{M}$ used in Algorithm 1 often includes the Jacobian of the forces (see Appendix B for the exact formulation). We denote it below as $\hat{\mathbf{M}}$ in order to mark the difference. A linear solve will be needed to compute the acceleration since it is time consuming to compute $\hat{\mathbf{M}}^{-1}$. We use implicit differentiation to compute the gradients of the linear solve. Given an equation $\hat{\mathbf{M}}\mathbf{a} = \mathbf{f}$ with a solution $\mathbf{z}$ and the propagated gradient $\frac{\partial \mathcal{L}}{\partial \mathbf{a}}|_{\mathbf{a}=\mathbf{z}}$, where $\mathcal{L}$ is the task-specific loss function, we can use the implicit differentiation form

$$\hat{\mathbf{M}}\partial\mathbf{a} = \partial\mathbf{f} - \partial\hat{\mathbf{M}}\mathbf{a} \tag{1}$$

to derive the gradient as

$$\frac{\partial \mathcal{L}}{\partial \hat{\mathbf{M}}} = -\mathbf{d}_{\mathbf{a}}\mathbf{z}^\top \quad \frac{\partial \mathcal{L}}{\partial \mathbf{f}} = \mathbf{d}_{\mathbf{a}}^\top, \tag{2}$$

where $\mathbf{d}_{\mathbf{a}}$ is obtained from the linear system

$$\hat{\mathbf{M}}^\top \mathbf{d}_{\mathbf{a}} = \frac{\partial \mathcal{L}}{\partial \mathbf{a}}^\top. \tag{3}$$

The proof is as follows. We take $\frac{\partial \mathcal{L}}{\partial \mathbf{f}}$ as an example here, the derivation of $\frac{\partial \mathcal{L}}{\partial \mathbf{M}}$ is shown in Appendix A.1:

$$\frac{\partial \mathcal{L}}{\partial \mathbf{f}} = \frac{\partial \mathcal{L}}{\partial \mathbf{a}} \cdot \frac{\partial \mathbf{a}}{\partial \mathbf{f}} = \mathbf{d}_{\mathbf{a}}^\top \hat{\mathbf{M}} \cdot \hat{\mathbf{M}}^\dagger \mathbf{I} = \mathbf{d}_{\mathbf{a}}^\top. \tag{4}$$

The first equality is given by the chain rule, the second is given by Equations 1 and 3, and $\hat{\mathbf{M}}^{\dagger}$ is the pseudoinverse of matrix $\hat{\mathbf{M}}$.

### 3.3 Dynamic Collision Detection and Response

As mentioned in Sec. 1, a static collision solver is not suitable for cloth because the total number of possible collision pairs is very high: quadratic in the number of faces. A common approach in cloth simulation is to dynamically detect collision on the fly and compute the response. We use a bounding volume hierarchy for collision detection [27], and non-rigid impact zones [12] to compute the collision response.

Specifically, we solve a cubic equation to detect the collision time $t$ of each vertex-face or edge-edge pair that is sufficiently close to contact:

$$(\mathbf{x}_1 + \mathbf{v}_1 t) \cdot (\mathbf{x}_2 + \mathbf{v}_2 t) \times (\mathbf{x}_3 + \mathbf{v}_3 t) = 0, \tag{5}$$

where $\mathbf{x}_k$ and $\mathbf{v}_k$ ($k = 1, 2, 3$) are the relative position and velocity to the first vertex. A solution that lies in $[0, 1]$ means that a collision is possible before the next simulation step. After making sure that the pair indeed intersects at time $t$, we set up one constraint for this collision, forcing the signed distance of this collision pair at time $t$ to be no less than the thickness of the cloth $\delta$. The signed distance of the vertex-face or edge-edge pair is linear to the vertex position $\mathbf{x}$. The set of all constraints then makes up a quadratic optimization problem as discussed later in Sec. 3.4.

For backpropagation, we need to compute the derivatives of the solution $t$ since it is related to the parameters of the constraints. We use implicit differentiation here to simplify the process. Generally, given a cubic equation $ax^3 + bx^2 + cx + d = 0$, its implicit differentiation is of the following form:

$$(3ax^2 + 2bx + c)\partial x = \partial a x^3 + \partial b x^2 + \partial c x + \partial d. \tag{6}$$

Therefore we have

$$\begin{bmatrix} \frac{\partial x}{\partial a} & \frac{\partial x}{\partial b} & \frac{\partial x}{\partial c} & \frac{\partial x}{\partial d} \end{bmatrix} = \frac{1}{3ax^2 + 2bx + c} \begin{bmatrix} x^3 & x^2 & x & 1 \end{bmatrix}. \tag{7}$$

### 3.4 Derivatives of the Collision Response

A general approach to integrating collision constraints into physics simulation has been proposed by Belbute-Peres *et al.* [7]. However, as mentioned in Sections 1 and 2, constructing a static LCP is often impractical in cloth simulation because of high dimensionality. Collisions that actually happen in each step are very sparse compared to the complete set. Therefore, we use a dynamic approach that incorporates collision detection and response.

Collision handling in our implementation is based on impact zone optimization [22]. It finds all colliding instances using continuous collision detection (Sec. 3.3) and sets up the constraints for all collisions. In order to introduce minimum change to the original mesh state, we develop a QP problem to solve for the constraints. Since the signed distance function is linear in $\mathbf{x}$, the optimization takes a quadratic form:

$$\underset{\mathbf{z}}{\text{minimize}} \quad \frac{1}{2}(\mathbf{z} - \mathbf{x})^{\top} \mathbf{W}(\mathbf{z} - \mathbf{x}) \tag{8}$$

$$\text{subject to} \quad \mathbf{G}\mathbf{z} + \mathbf{h} \leq \mathbf{0} \tag{9}$$

where $\mathbf{W}$ is a constant diagonal weight matrix related to the mass of each vertex, and $\mathbf{G}$ and $\mathbf{h}$ are constraint parameters (see Appendix B for more details). We further denote the number of variables and constraints by $n$ and $m$, *i.e.* $\mathbf{x} \in \mathbb{R}^n$, $\mathbf{h} \in \mathbb{R}^m$, and $\mathbf{G} \in \mathbb{R}^{m \times n}$. Note that this optimization is a function with inputs $\mathbf{x}$, $\mathbf{G}$, and $\mathbf{h}$, and output $\mathbf{z}$. Our goal here is to derive $\frac{\partial \mathcal{L}}{\partial \mathbf{x}}$, $\frac{\partial \mathcal{L}}{\partial \mathbf{G}}$, and $\frac{\partial \mathcal{L}}{\partial \mathbf{h}}$ given $\frac{\partial \mathcal{L}}{\partial \mathbf{z}}$, where $\mathcal{L}$ refers to the loss function.

When computing the gradient using implicit differentiation [1], the dimensionality of the linear system (Equation 13) can be too high. Our key observation here is that $n \gg m > \text{rank}(\mathbf{G})$, since one contact often involves 4 vertices (thus 12 variables) and some contacts may be linearly dependent (*e.g.* multiple adjacent collision pairs). OptNet [1] solves a linear equation of size $m + n$, which is more than necessary. We introduce a simpler and more efficient algorithm below to minimize the size of the linear equation.

### 3.4.1 QR Decomposition

To make things simpler, we assume that $\mathbf{G}$ is of full rank in this section. At global minimum $\mathbf{z}^*$ and $\lambda^*$ of the Lagrangian, the following holds for stationarity and complementary slackness conditions:

$$\mathbf{W}\mathbf{z}^* - \mathbf{W}\mathbf{x} + \mathbf{G}^\top \lambda^* = 0 \tag{10}$$

$$D(\lambda^*)(\mathbf{G}\mathbf{z}^* + \mathbf{h}) = 0, \tag{11}$$

with their implicit differentiation as

$$\begin{bmatrix} \mathbf{W} & \mathbf{G}^\top \\ D(\lambda^*)\mathbf{G} & D(\mathbf{G}\mathbf{z}^* + \mathbf{h}) \end{bmatrix} \begin{bmatrix} \partial \mathbf{z} \\ \partial \lambda \end{bmatrix} = \begin{bmatrix} \mathbf{M}\partial \mathbf{x} - \partial \mathbf{G}^\top \lambda^* \\ -D(\lambda^*)(\partial \mathbf{G}\mathbf{z}^* + \partial \mathbf{h}) \end{bmatrix}, \tag{12}$$

where $D()$ transforms a vector to a diagonal matrix. Using similar derivation to Sec. 3.2, solving the equation

$$\begin{bmatrix} \mathbf{W} & \mathbf{G}^\top D(\lambda^*) \\ \mathbf{G} & D(\mathbf{G}\mathbf{z}^* + \mathbf{h}) \end{bmatrix} \begin{bmatrix} \mathbf{d_z} \\ \mathbf{d_\lambda} \end{bmatrix} = \begin{bmatrix} \frac{\partial \mathcal{L}}{\partial \mathbf{z}}^\top \\ \mathbf{0} \end{bmatrix} \tag{13}$$

can provide the desired gradient:

$$\frac{\partial \mathcal{L}}{\partial \mathbf{x}} = \mathbf{d}_{\mathbf{z}}^T \mathbf{W} \tag{14}$$

$$\frac{\partial \mathcal{L}}{\partial \mathbf{G}} = -D(\lambda^*)\mathbf{d}_\lambda \mathbf{z}^{*\top} - \lambda^* \mathbf{d}_{\mathbf{z}}^\top \tag{15}$$

$$\frac{\partial \mathcal{L}}{\partial \mathbf{h}} = -\mathbf{d}_\lambda^T D(\lambda^*). \tag{16}$$

(See Appendix A.2 for the derivation.) However, as mentioned before, directly solving Equation 13 may be computationally expensive in our case. We show that by performing a QR decomposition, the solution can be derived without solving a large system.

To further reduce computation, we assume that no constraint is 'over-satisfied', *i.e.* $\mathbf{G}\mathbf{z}^* + \mathbf{h} = \mathbf{0}$. We will remove these assumptions later in Sec. 3.4.2. We compute the QR decomposition of $\sqrt{\mathbf{W}}^{-1}\mathbf{G}^\top$:

$$\sqrt{\mathbf{W}}^{-1}\mathbf{G}^\top = \mathbf{Q}\mathbf{R}. \tag{17}$$

The solution of Equation 13 can be expressed as

$$\mathbf{d_z} = \sqrt{\mathbf{W}}^{-1}(\mathbf{I} - \mathbf{Q}\mathbf{Q}^\top)\sqrt{\mathbf{W}}^{-1}\frac{\partial \mathcal{L}}{\partial \mathbf{z}}^\top \tag{18}$$

$$\mathbf{d}_\lambda = D(\lambda^*)^{-1}\mathbf{R}^{-1}\mathbf{Q}^\top \sqrt{\mathbf{W}}^{-1}\frac{\partial \mathcal{L}}{\partial \mathbf{z}}^\top , \tag{19}$$

where $\sqrt{\mathbf{W}}^{-1}$ is the inverse of the square root of a diagonal matrix. The result above can be verified by substitution in Equation 13.

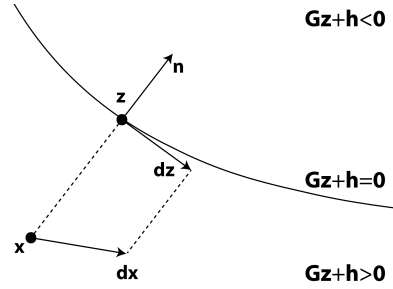

Figure 1: Impact of perturbation. A small perturbation of the target position will cause the final result to move along the constraint surface.

The intuition behind Equation 18 is as follows. When perturbing the original point $\mathbf{x}$ in an optimization, the resulting displacement of $\mathbf{z}$ will be moving along the surface of $\mathbf{G}\mathbf{x} + \mathbf{h} = 0$, which will become perpendicular to the normal when the perturbation is small. (Fig. 1 illustrates this idea in two dimensions.) This is where the term $\mathbf{I} - \mathbf{Q}\mathbf{Q}^\top$ comes from. Note that $\sqrt{\mathbf{W}}^{-1}\mathbf{G}^\top$ is an $n \times m$ matrix, where $n \gg m$ and the QR decomposition will only take $O(nm^2)$ time, compared to $O((n+m)^3)$ in the original dense linear solve. After that we will need to solve a linear equation in Equation 19, but it is more efficient than solving Equation 13 since it is only of size $m$, and $\mathbf{R}$ is an upper-triangular matrix. In our collision response case, where $n \leq 12m$, our method can provide up to 183x acceleration in theory. The speed-up in our experiments (Sec. 4) ranges from 60x to 130x for large linear systems.

### 3.4.2 Low-rank Constraints

The algorithm above cannot be directly applied when $\mathbf{G}$ is low-rank, or when some constraint is not at boundary. This will cause $\mathbf{R}$ or $D(\lambda^*)$ to be singular. We now show that the singularity can be avoided via small modifications to the algorithm.

First, if $\lambda_k = 0$ for the $k^{\text{th}}$ constraint then $\mathbf{d}\lambda_k$ doesn't matter. This is because the final result contains only components of $D(\lambda^*)\mathbf{d}\lambda$ but not $\mathbf{d}\lambda$ alone, as shown in Equations 15 and 16. Intuitively, if the constraint is over-satisfied, then perturbing the parameters of that constraint will not have impact on $\mathbf{z}$. Based on this observation, we can remove the constraints in $\mathbf{G}$ when their corresponding $\lambda$ is 0.

Next, if $\mathbf{G}$ is of rank $k$, where $k < m$, then we can rewrite Equation 17 as

$$\sqrt{\mathbf{W}}^{-1}\mathbf{G}^\top = \mathbf{Q}_1[\mathbf{R}_1 \quad \mathbf{R}_2], \tag{20}$$

where $\mathbf{Q}_1 \in \mathbb{R}^{n \times k}$, $\mathbf{R}_1 \in \mathbb{R}^{k \times k}$, and $\mathbf{R}_2 \in \mathbb{R}^{k \times (m-k)}$. Getting rid of $\mathbf{R}_2$ (*i.e.* removing those constraints from the beginning) does not affect the optimization result, but may change $\lambda$ so that the computed gradients are incorrect. Therefore, we need to transfer the Lagrange multipliers to the linearly independent terms first:

$$\lambda_1 \leftarrow \lambda_1 + \mathbf{R}_1^{-1}\mathbf{R}_2\lambda_2, \tag{21}$$

where $\lambda_1$ and $\lambda_2$ are the Lagrange multipliers corresponding to the constraints on $\mathbf{R}_1$ and $\mathbf{R}_2$.

## 4 Experiments

We conduct three experiments to showcase the power of differentiable cloth simulation. First, we use an ablation study to quantify the performance gained by using our method to compute the gradient. Next, we use the computed gradient to optimize the physical parameters of cloth. Lastly, we demonstrate the ability to control cloth motion.

### 4.1 Ablation Study

As mentioned in Sec. 3.4.1, our method for computing the gradients of the optimization can achieve a speed-up of up to 183x in theory. We conduct an ablation study to verify this estimate in practice. In order to clearly measure the timing difference, we design a scenario with many collisions. We put a piece of cloth into an upside-down square pyramid, so that the cloth is forced to fold, come into frequent contact with the pyramid, and collide with itself, as shown in Fig. 2.

We measure the running time of backpropagation in each quadratic optimization and also the running time of the physics solve as a reference. With all other variables fixed, we compare to the baseline method where the gradients are computed by directly solving Equation 13. Timings are listed in Tab. 1. In this experiment, the backpropagation of the physics solve takes from 0.007s to 0.5s, which, together with the timings of the baseline, implies that the collision handling step is the critical bottleneck when there are many collisions in the scene. The results in Tab. 1 show that our proposed method can significantly decrease the matrix size required for computation and thus the actual running time, resolving the bottleneck in backpropagation.

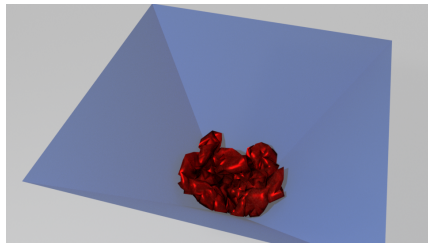

Figure 2: Example frame from the ablation study. A piece of cloth is crumpled inside a square pyramid, so as to generate a large number of collisions.

The experimental results also match well with the theory in Sec. 3.4. Each collision involves a vertex-face or edge-edge pair, which both have 4 vertices and 12 variables. Therefore, the original matrix size ($n + m = 13m$) should be about 13 times bigger than in our method ($m$). In our experiment, the ratio of the matrix size is indeed close to 13. Possible reasons for the ratio not being exactly 13 include (a) multiple collision pairs that share the same vertex, making $n$ smaller, and (b) the constraint matrix can be of low rank, as described in Sec. 3.4.2, making the effective $m$ smaller in practice.

| Mesh resolution | Baseline | | Ours | | Speedup | |
|---|---|---|---|---|---|---|
| | Matrix size | Runtime (s) | Matrix size | Runtime (s) | Matrix size | Runtime |
| 16x16 | $599 \pm 76$ | $0.33 \pm 0.13$ | $\mathbf{66 \pm 26}$ | $\mathbf{0.013 \pm 0.0019}$ | 8.9 | 25 |
| 32x32 | $1326 \pm 23$ | $1.2 \pm 0.10$ | $\mathbf{97 \pm 24}$ | $\mathbf{0.011 \pm 0.0023}$ | 13 | 112 |
| 64x64 | $2024 \pm 274$ | $4.6 \pm 0.33$ | $\mathbf{242 \pm 47}$ | $\mathbf{0.072 \pm 0.011}$ | 8.3 | 64 |

Table 1: Statistics of the backward propagation with and without our method for various mesh resolutions. We report the average values in each cell with the corresponding standard deviations. By using our method, the runtime of gradient computation is reduced by up to two orders of magnitude.

## 4.2 Material Estimation

In this experiment, our aim is to learn the material parameters of cloth from observation. The scene features a piece of cloth hanging under gravity and subjected to a constant wind force, as shown in Fig. 3. We use the material model from Wang *et al.* [30]. It consists of three parts: density $d$, stretching stiffness $\mathbf{S}$, and bending stiffness $\mathbf{B}$. The stretching stiffness quantifies how large the reaction force will be when the cloth is stretched out; the bending stiffness models how easily the cloth can be bent and folded.

We used the real-world dataset from Wang *et al.* [30], which consists of 10 different cloth materials. There are in total 50 frames of simulated data. The first 25 frames are taken as input and all 50 frames are used to measure accuracy. This is a case-by-case optimization problem. Our goal is to fit the observed data in each sequence as well as possible, with no "training set" used for training.

In our optimization setup, we use SGD with learning rate ranging from 0.01 to 0.1 and momentum from 0.9 to 0.99, depending on the convergence speed. The initial guess is the set of average values across all materials. We define the loss as the average MSE across all frames. In order to speed up optimization, we gradually increase the number of frames used. Specifically, we first optimize the parameters using only 1 simulated frame. We proceed to the second frame after the loss decreases to a certain threshold. This optimization scheme can help obtain a relatively good guess before additional frames are involved.

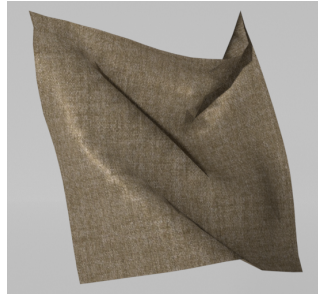

Figure 3: Example frame from the material estimation scene for cloth blowing in the wind.

As a simple baseline, we measure the total external force and divide it by the observed acceleration to compute the density. For the stretching stiffness, we simplify the model to an isotropic one and record the maximum deformation magnitude along the vertical axis. Since the effect of the bending stiffness is too subtle to observe, we directly use the averaged value as our prior. We also compare our method with the L-BFGS optimization by Wang *et al.* [30] using finite difference. We used the PyTorch L-BFGS implementation and set the learning rate ranging from 0.1 to 0.2 depending on the convergence speed.

For the performance measurement, we use the Frobenius norm normalized by the target as the metric for the material parameters:

$$\mathcal{E}(\mathbf{P}) = \frac{\|\mathbf{P} - \mathbf{P}_0\|_F}{\|\mathbf{P}_0\|_F}, \tag{22}$$

where $\mathbf{P}$ and $\mathbf{P}_0$ are the estimated and the target physics parameters, which stand for either density $d$, stretching stiffness $\mathbf{S}$, or bending stiffness $\mathbf{B}$. In order to show the final visual effect, we also measure the average distance of the vertices between the estimated one and the target normalized by the size of the cloth as another metric:

$$\mathcal{E}(\mathbf{X}) = \frac{1}{nTL} \sum_{1 \leq i \leq T, 1 \leq j \leq n} \|\mathbf{X}_{i,j} - \mathbf{Y}_{i,j}\|_2, \tag{23}$$

where $L$ is the size of the cloth, and $\mathbf{X}$ and $\mathbf{Y}$ are $T \times n \times 3$ matrices denoting the $n$ simulated vertex positions across $T$ frames using the estimated parameter and the target, respectively.

Tab. 2 shows the estimation result. We achieve a much smaller error in most measurements in comparison to the baselines. The reason the stiffness matrices do not have low error is that (a) a large

part of them describes the nonlinear stress behavior that needs a large deformation of the cloth and is not commonly observed in our environment, (b) different stiffness values can sometimes provide similar results, and (c) the bending force for common cloth materials is too small compared to gravity and the wind forces to make an impact. The table shows that the linear part of the stiffness matrix is optimized well. With the computed gradient using our model, one can effectively optimize the unknown parameters that dominate the cloth movement to fit the observed data. We show in the supplementary video that the estimated parameters yield very similar qualitative behavior to the original observation.

Compared with regular simulators, our simulator is designed to be embedded in deep networks. When gradients are needed, our simulator shows significant improvement over finite-difference methods, as shown in Tab. 2. Regular simulators need to run one simulation for each input variable to compute the gradient, while our method only needs to run once for all gradients to be computed. Therefore, the more input variables there are during learning, the greater the performance gain that can be achieved by our method over finite-difference methods.

| Method | Runtime (sec/step/iter) | Density error (%) | Non-ln streching stiffness error (%) | Ln streching stiffness error (%) | Bending stiffness error (%) | Simulation error (%) |
|---|---|---|---|---|---|---|
| Baseline | - | $68 \pm 46$ | $74 \pm 23$ | $160 \pm 119$ | $\mathbf{70 \pm 42}$ | $12 \pm 3.0$ |
| L-BFGS [30] | $2.89 \pm 0.02$ | $4.2 \pm 5.6$ | $64 \pm 34$ | $72 \pm 90$ | $70 \pm 43$ | $4.9 \pm 3.3$ |
| Ours | $\mathbf{2.03 \pm 0.06}$ | $\mathbf{1.8 \pm 2.0}$ | $\mathbf{57 \pm 29}$ | $\mathbf{45 \pm 41}$ | $77 \pm 36$ | $\mathbf{1.6 \pm 1.4}$ |

Table 2: Results on the material parameter estimation task. Lower is better. 'Ln' stands for 'linear'. Values of the material parameters are the Frobenius norms of the difference normalized by the Frobenius norm of the target. Values of the simulated result are the average pairwise vertex distance normalized by the size of the cloth. Our gradient-based method yields much smaller error than the baselines.

## 4.3 Motion Control

We further demonstrate the power of our differentiable simulator by optimizing control parameters. The task is to drop a piece of cloth into a basket, as shown in Fig. 4. The cloth is originally placed on a table that is away from the basket. The system then applies external forces to the corners of the cloth to lift it and drop it into the basket. The external force is applied for 3 seconds and can be changed during this period. The basket is a box with an open top. A planar obstacle is placed between the cloth and the basket to increase the difficulty of the task.

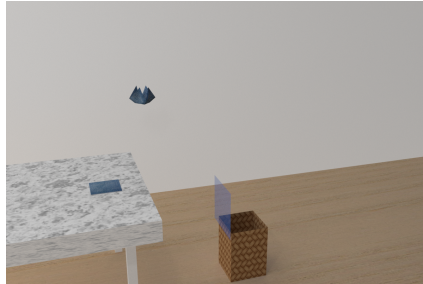

Figure 4: Example frame from the motion control experiment: dropping cloth into a basket.

We define the loss here as the squared distance between the center of mass of the cloth and the bottom of the basket. To demonstrate the ability to embed the simulator into neural networks, we also couple our simulator with a two-layer fully-connected (FC) network that takes the mesh states as input and outputs the control forces. Our methods here are compared to two baselines. One of the baselines is a simple method that computes the momentum needed at every time step. The entire cloth is treated as a point mass and an external force is computed at each time step to control the point mass towards the goal. Obstacles are simply neglected in this method. The other baseline is the PPO algorithm, as implemented in Ray RLlib [18]. The reward function is defined as the negative of the distance of the center of mass of the cloth to the bottom of the basket. Please refer to the Appendix for additional details.

Tab. 3 shows the performance of the different methods and their sample complexity. The error shown in the table is the distance defined above normalized by the size of the cloth. Our method achieves the best performance with a much smaller number of simulation steps. The bottom of the basket in our setting has the same size as the cloth, so a normalized error of less than 50%, as our methods achieve, implies that the cloth is successfully dropped into the basket.

| Method | Error (%) | Samples |
|---|---|---|
| Point mass | 111 | – |
| PPO [18] | 432 | 10,000 |
| Ours | **17** | **53** |
| Ours+FC | 39 | 108 |

Table 3: Motion control results. The table reports the smallest distance to the target position, normalized by the size of the cloth, and the number of samples used during training.

## 5    Conclusion

We presented a differentiable cloth simulator that can compute the analytical gradient of the simulation function with respect to the input parameters. We used dynamic collision handling and explicitly derived its gradient. Implicit differentiation is used in computing gradients of the linear solver and collision response. Experiments have demonstrated that our method accelerates backpropagation by up to two orders of magnitude.

We have demonstrated the potential of differentiable cloth simulation in two application scenarios: material estimation and motion control. By making use of the gradients from the physically-aware simulation, our method can optimize the unknown parameters faster and more accurately than gradient-free baselines. Using differentiable simulation, we can learn the intrinsic properties of cloth from observation.

One limitation of our existing implementation is that the current simulation architecture is not optimized for large-scale vectorized operations, which introduces some overhead. This can be addressed by a specialized, optimized simulation system based solely on tensor operations.

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
