[Supplementary Material]

# Appendix A Derivations of the Gradient Computation

## A.1 Proof of Equation 2

We now show that $\frac{\partial \mathcal{L}}{\partial \hat{\mathbf{M}}} = -\mathbf{d_a z}^\top$. For convenience of expression, we split the matrix $\hat{\mathbf{M}}$ into elements $\{\hat{\mathbf{M}}_{i,j}\}$. Setting irrelevant variables to zero, we obtain from Equation 1 that:

$$\hat{\mathbf{M}} \partial \mathbf{a} = -\partial \hat{\mathbf{M}} \mathbf{z} = \begin{pmatrix} \mathbf{0} \\ -\partial \hat{\mathbf{M}}_{i,j} \mathbf{z}_j \\ \mathbf{0} \end{pmatrix} \tag{24}$$

Hence, we have:

$$\frac{\partial \mathbf{a}}{\partial \hat{\mathbf{M}}_{i,j}} = \hat{\mathbf{M}}^\dagger \begin{pmatrix} \mathbf{0} \\ -\mathbf{z}_j \\ \mathbf{0} \end{pmatrix} \tag{25}$$

Similar to Equation 4, we arrive at:

$$\frac{\partial \mathcal{L}}{\partial \hat{\mathbf{M}}_{i,j}} = \frac{\partial \mathcal{L}}{\partial \mathbf{a}} \cdot \frac{\partial \mathbf{a}}{\partial \hat{\mathbf{M}}_{i,j}} = \mathbf{d_a}^\top \hat{\mathbf{M}} \cdot \hat{\mathbf{M}}^\dagger \begin{pmatrix} \mathbf{0} \\ -\mathbf{z}_j \\ \mathbf{0} \end{pmatrix} = -\mathbf{d}_{\mathbf{a}_i} \mathbf{z}_j \tag{26}$$

Combining all elements in $\hat{\mathbf{M}}$ together we have:

$$\frac{\partial \mathcal{L}}{\partial \hat{\mathbf{M}}} = -\mathbf{d_a z}^\top \tag{27}$$

## A.2 Proof of Equation 13-16

Let $\hat{\mathbf{z}} = \begin{bmatrix} \mathbf{z} & \lambda \end{bmatrix}^\top$. Using Equation 12 we have:

$$\frac{\partial \hat{\mathbf{z}}}{\partial \mathbf{x}} = \begin{bmatrix} \mathbf{W} & \mathbf{G}^\top \\ D(\lambda^*)\mathbf{G} & D(\mathbf{Gz}^* + \mathbf{h}) \end{bmatrix}^\dagger \begin{bmatrix} \mathbf{W} \\ \mathbf{0} \end{bmatrix} \tag{28}$$

$$\frac{\partial \hat{\mathbf{z}}}{\partial \mathbf{h}} = \begin{bmatrix} \mathbf{W} & \mathbf{G}^\top \\ D(\lambda^*)\mathbf{G} & D(\mathbf{Gz}^* + \mathbf{h}) \end{bmatrix}^\dagger \begin{bmatrix} \mathbf{0} \\ -D(\lambda^*) \end{bmatrix} \tag{29}$$

Then, the chain rule can yield the results as:

$$\frac{\partial \mathcal{L}}{\partial \mathbf{x}} = \frac{\partial \mathcal{L}}{\partial \hat{\mathbf{z}}} \cdot \frac{\partial \hat{\mathbf{z}}}{\partial \mathbf{x}} \tag{30}$$

$$= \begin{bmatrix} \mathbf{d_z}^\top & \mathbf{d_\lambda}^\top \end{bmatrix} \begin{bmatrix} \mathbf{W} & \mathbf{G}^\top \\ D(\lambda^*)\mathbf{G} & D(\mathbf{Gz}^* + \mathbf{h}) \end{bmatrix} \cdot \begin{bmatrix} \mathbf{W} & \mathbf{G}^\top \\ D(\lambda^*)\mathbf{G} & D(\mathbf{Gz}^* + \mathbf{h}) \end{bmatrix}^\dagger \begin{bmatrix} \mathbf{W} \\ \mathbf{0} \end{bmatrix} \tag{31}$$

$$= \mathbf{d_z}^\top \mathbf{W} \tag{32}$$

$$\frac{\partial \mathcal{L}}{\partial \mathbf{h}} = \frac{\partial \mathcal{L}}{\partial \hat{\mathbf{z}}} \cdot \frac{\partial \hat{\mathbf{z}}}{\partial \mathbf{h}} \tag{33}$$

$$= \begin{bmatrix} \mathbf{d_z}^\top & \mathbf{d_\lambda}^\top \end{bmatrix} \begin{bmatrix} \mathbf{W} & \mathbf{G}^\top \\ D(\lambda^*)\mathbf{G} & D(\mathbf{Gz}^* + \mathbf{h}) \end{bmatrix} \cdot \begin{bmatrix} \mathbf{W} & \mathbf{G}^\top \\ D(\lambda^*)\mathbf{G} & D(\mathbf{Gz}^* + \mathbf{h}) \end{bmatrix}^\dagger \begin{bmatrix} \mathbf{0} \\ -D(\lambda^*) \end{bmatrix} \tag{34}$$

$$= -\mathbf{d_\lambda}^\top D(\lambda^*) \tag{35}$$

Similarly as Appendix A.1, we split the matrix $\mathbf{G}$ into elements $\{\mathbf{G}_{i,j}\}$. From Equation 12 we have:

$$\begin{bmatrix} \mathbf{W} & \mathbf{G}^\top \\ D(\lambda^*)\mathbf{G} & D(\mathbf{Gz}^* + \mathbf{h}) \end{bmatrix} \partial \hat{\mathbf{z}} = \begin{bmatrix} -\partial \mathbf{G}^\top \lambda^* \\ -D(\lambda^*)\partial \mathbf{Gz}^* \end{bmatrix} = \begin{bmatrix} \mathbf{0} \\ -\partial \mathbf{G}_{i,j} \lambda_i^* \\ \mathbf{0} \\ -\lambda_i^* \partial \mathbf{G}_{i,j} \mathbf{z}_j^* \\ \mathbf{0} \end{bmatrix} \tag{36}$$

which indicates that:

$$\frac{\partial \hat{\mathbf{z}}}{\partial \mathbf{G}_{i,j}} = \begin{bmatrix} \mathbf{W} & \mathbf{G}^\top \\ D(\lambda^*)\mathbf{G} & D(\mathbf{G}\mathbf{z}^* + \mathbf{h}) \end{bmatrix}^\dagger \begin{bmatrix} \mathbf{0} \\ -\lambda_i^* \\ \mathbf{0} \\ -\lambda_i^* \mathbf{z}_j^* \\ \mathbf{0} \end{bmatrix} \tag{37}$$

So the chain rule gives:

$$\frac{\partial \mathcal{L}}{\partial \mathbf{G}_{i,j}} = \frac{\partial \mathcal{L}}{\partial \hat{\mathbf{z}}} \cdot \frac{\partial \hat{\mathbf{z}}}{\partial \mathbf{G}_{i,j}} \tag{38}$$

$$= \begin{bmatrix} \mathbf{d}_\mathbf{z}^\top & \mathbf{d}_\lambda^\top \end{bmatrix} \begin{bmatrix} \mathbf{W} & \mathbf{G}^\top \\ D(\lambda^*)\mathbf{G} & D(\mathbf{G}\mathbf{z}^* + \mathbf{h}) \end{bmatrix} \cdot \begin{bmatrix} \mathbf{W} & \mathbf{G}^\top \\ D(\lambda^*)\mathbf{G} & D(\mathbf{G}\mathbf{z}^* + \mathbf{h}) \end{bmatrix}^\dagger \begin{bmatrix} \mathbf{0} \\ -\lambda_i^* \\ \mathbf{0} \\ -\lambda_i^* \mathbf{z}_j^* \\ \mathbf{0} \end{bmatrix} \tag{39}$$

$$= -\mathbf{d}_{\mathbf{z}_j} \lambda_i^* - \mathbf{d}_{\lambda_i} \lambda_i^* \mathbf{z}_j^* \tag{40}$$

Combining all elements in $\mathbf{G}$ together, we have:

$$\frac{\partial \mathcal{L}}{\partial \mathbf{G}} = -\lambda^* \mathbf{d}_\mathbf{z}^\top - D(\lambda^*)\mathbf{d}_\lambda \mathbf{z}^{*\top} \tag{41}$$

## Appendix B   Cloth Simulation Basics

Generally, cloth simulation includes three steps: force computation, dynamic solve, and collision handling. Extra steps, such as plasticity handling and strain limiting, are omitted since they are not essential components of a basic cloth simulation.

### B.1   Force Computation

For external forces, the most common ones are gravity and wind forces, which are both straightforward. We focus on internal, constraint and frictional forces here.

Clothes are usually modeled as a 2D manifold mesh in 3D space. We apply Finite Element Method (FEM) to compute internal forces. For each triangle face in the mesh, we compute the deformation gradient as a variable of the strain:

$$\mathbf{F} = \frac{\partial \mathbf{x}}{\partial \mathbf{X}} \tag{42}$$

Here, $\mathbf{x}$ is the current 3D position of the triangle, and $\mathbf{X}$ is their coordinate in the 2D material space. Then, the stress (or internal forces) is computed using the deformation gradient $\mathbf{F}$. Usually a strain energy $\mathbf{E}$ is defined and we use its negative gradient as the force. In our base simulator, the stress is defined as a piece-wise linear function regarding the Green-Lagrange Strain, defined by Wang *et al.* [30]:

$$\mathbf{E} = \frac{1}{2}(\mathbf{F}^\top \mathbf{F} - \mathbf{I}) \tag{43}$$

Note that due to the geometric modeling of the cloth, there is no force caused by the thickness of the cloth. Most simulators use an extra 'bending force' as a compensation, following Grinspun *et al.* [10]. The bending force is defined between two adjacent faces when their dihedral angle is not a resting one.

The other two categories are relatively simpler. Constraint forces are defined as the negative gradient of the constraint energy, while frictional forces are created when two objects are in close proximity and have relative motions.

### B.2   Dynamic Solve

In the simplest case, we solve $\mathbf{M}\mathbf{a} = \mathbf{f}$ for the acceleration and update the position and velocity accordingly, as shown in Algorithm 1. This Forward Euler method suffers from the well-known

stability issue and often limits the time step size for the simulation. In order to take larger step for faster simulation, Backward Euler is often used. More specifically, we want our acceleration to match the force computed in the next time step:

$$\mathbf{M}\frac{\Delta \mathbf{v}}{\Delta t} = \mathbf{f}(\mathbf{x} + \Delta \mathbf{x}) = \mathbf{f}(\mathbf{x} + \Delta t(\mathbf{v} + \Delta \mathbf{v})) \tag{44}$$

By using Taylor Expansion, we have:

$$(\mathbf{M} - \Delta t^2 \frac{\partial \mathbf{f}}{\partial \mathbf{x}})\Delta \mathbf{v} = \Delta t \mathbf{f}(\mathbf{x} + \Delta t \mathbf{v}) \tag{45}$$

So the matrix used in the linear solve (Sec. 3.2) is defined as:

$$\hat{\mathbf{M}} = \mathbf{M} - \Delta t^2 \frac{\partial \mathbf{f}}{\partial \mathbf{x}} \tag{46}$$

As long as we have the Jacobian of the forces $\frac{\partial \mathbf{f}}{\partial \mathbf{x}}$, we can compute a more stable result of $\Delta \mathbf{v}$ and can apply larger $\Delta t$, as discussed by Baraff and Witkin [2].

### B.3 Collision Handling

As introduced in Sec. 3.3, we used continuous collision detection between two simulation steps to detect all possible collisions. When two faces collide with each other, there are two different collision types: vertex-face collision and edge-edge collision. The common trait is that at time of collision, the four involved vertices are in the same plane. Based on this, we can develop and solve a cubic equation regarding the time of collision, $t$ (Sec. 3.3).

When the collision is detected, we need to form the corresponding constraint at time $t$:

$$\left(\sum_{k=1}^{4} w_k \mathbf{x}_k(t)\right) \cdot \mathbf{n} \geq d \tag{47}$$

Here, $w_k$ is the weight parameter, $\mathbf{x}_k(t)$ is the vertex position at time $t$, $\mathbf{n}$ is the normal of the plane, and $d$ is the cloth thickness. The weight parameters are determined using barycentric coordinates of the intersection point in the face (in the vertex-face collision case) or on the edges (in the edge-edge collision case).

We consider $w$ and $\mathbf{n}$ as constants during the optimization, and $\mathbf{x}_k(t)$ is linearly interpolated between two time steps. So it is a linear constraint regarding $\mathbf{x}$. Combining all constraints together, we have:

$$\mathbf{G}\mathbf{x} + \mathbf{h} \leq 0 \tag{48}$$

as shown in Equation 9.

In the collision response phase, we want to introduce minimum energy to move the vertex away so that all constraints can be satisfied. Therefore, we form this optimization as a QP problem, as shown in the main text (Sec. 3.4). This is also why the objective function is weighted by the mass.

Figure 5: A motion control scene with more obstacles. The cloth needs to drop down and slide through the slopes to get to the target position.

## Appendix C    Characterization of Control Task

The initial control force is set to zero. The control network consists of two FC layers, where the input (size $81 \times 2 \times 3$) is the position and velocity of each vertex, the hidden layer is of size 200, and the output is the control force (size $4 \times 3$). The learning rate is $10^{-4}$ and the momentum is $0.5$. The reported result is the best among 10 trials.

## Appendix D    Collision-rich Motion Control

We here demonstrate an example of motion control application with richer collisions. As shown in Figure 5, there is a series of obstacles above the basket that preclude the cloth from falling directly into it. The variable settings are the same as described in Sec. 4.3. Our differentiable simulation provides the task with correct gradients so that the cloth is deposited into the basket.