[Reviews · NeurIPS 2019]

Reviewer 1



This submission fits into a recent trend of developing more expressive, "layers" for deep learning, with the intent of adding more structure to data-driven models. A pivotal part of achieving these goals is making such layers differentiable in an efficient manner. Here, the authors develop a differentiable cloth simulator. Moreover, the authors also improve on a previously developed method for differentiating through optimization problems in order to achieve a significantly more efficient collision resolution. This paper takes ideas from existing works (such as differentiable optimization, differentiable physics simulations), applies them to a new domain and employ a previously unused factorization method to arrive at an original contribution. Also as a positive note for such a practical contribution, the authors have committed to releasing the source code openly. The paper is well written and organized. The concepts are exposed clearly and a good level of detail is provided. Nevertheless, the paper would probably benefit subtly by having a supplement with some extra details, such as: - The derivations mentioned but not written out in lines 136 and 163 - Full characterization of control task: what random initialization is performed? what are the specifics of the control network? are the reported results an average of N trials (or the best)? - Comparisons to other simulators (eg non differentiable ones), or to numerical methods (finite differences) - How are the interaction between the cloth simulation and the rigid bodies in the experimental scenes dealt with? Could this type of cloth simulation be easily integrated into broader (differentiable) physics simulations? The experiments performed demonstrate convincingly the strengths of the proposed method in a variety of settings. As mentioned before, for the sake of completeness, I would be interested in also seeing a comparison of the proposed simulator with numerical differentiation, if feasible for the problem sizes. Also, given significant efficiency gains in this method, it would be interesting to see the comparison of the differentiable simulator to a regular one.

Reviewer 2



This paper presents a differentiable cloth simulation method. It formulates cloth simulation as a QP with collision handling as constraints and it follows [1][7] to compute derivatives of linear systems of KKT conditions, but finds the high-dimensionality of cloth simulation leads to a impractical solution. --Originality The novelty of this paper lies in that the authors uses dynamic collision detection to figure out a sparse size of constraints so that a QR decomposition is used to solve linear systems of KKT conditions efficiently, which makes it practical to use in optimization and learning. --Quality The references are sufficient and evaluations are compelling. --Clarity The notations can be improved. In Algorithm 1, M, f and a are not explained. In Line 117, it is better to say explicitly k = 1,2,3. The whole description of differentiable cloth simulation seems to fall in parts. It is not clear to me how Algorithm 1 corresponds to the QP problem. Section 3.2 seems to be used to give an example of how one collision is handled, but its connection to the matrix formulation used later doesn't seem to be obvious. In Section 3.3, it will be great to use exact (physically meaningful) notations to describe the linear system in stead of using general Ax=b. The G and h also lacks of explanations of their physical meanings in Section 3.4. --Significance The reviewer finds this work tackles an important problem in deep learning and provides a practical solution to it. The public release of its code will further improve the impact of this paper.

Reviewer 3



Devising differentiable physics simulation has recently attracted many interests due to its potential to be combined with deep learning algorithms. This paper introduces a new type of differentiable physics simulation, specifically designed for cloth simulation. The key challenge in making cloth simulation differentiable is due to the complexity in handling external and self collisions. The paper proposed to solve this problem by formulating the collision handling as a constrained optimization problem and derived its derivatives using implicit differentiation. Furthermore, they leveraged the thin shape of the constraint matrix and used QR decomposition to accelerate the computation of the gradients. They evaluated the method on two applications: material estimation and motion control. The result shows that the proposed method can successfully achieve the given task and is more efficient than the baseline methods. The paper is well written and in general clear. I do have a few comments about the proposed method, as listed below. 1. For the material estimation experiment, the real-world data from Wang et al. was used. I’m wondering how was the target physical parameters obtained? Since Wang et al. also went through an optimization process to obtain the simulation model that fits the real-world data, it seems more suitable to compare the proposed gradient-based method to Wang et al. directly in terms of the per-vertex error, instead of the simplified baseline. 2. The main challenge and contribution of the paper in my opinion is about collision handling in cloth simulation. However, the examples shown in the video are fairly sparse in collisions. The ablation study in 4.1 is more collision rich, but wasn’t used to optimize anything. It would be desired to see examples of material estimation or motion control that are collision rich. 3. How well does the method handle frictional contact? It is not discussed in the paper and is not reflected in the experiments. ######################################## After reading the authors' rebuttal and other reviews, most of my concerns have been addressed. Thus I will raise my score for the submission.

[Author Response · NeurIPS 2019]

*Dear Reviewers*: Thank you for the comments. We address the main issues and clarify some confusions below.

**Comparison to optimization methods (e.g., Wang et al.) using finite differences (Reviewers #1, #3).** To obtain the
ground-truth stretching and bending parameters, Wang et al. designed a number of controlled real-world environments.
With known external forces and labeled data, they used L-BFGS to optimize the parameters to fit the observed data.
They used finite differences to estimate the gradient.

For comparison, we run their optimization method in our environments, as requested. We used the PyTorch L-BFGS
implementation and set the learning rate ranging from 0.1 to 0.2 depending on the convergence speed. Using the best
parameters that we can obtain, we report runtime and accuracy in Table 1. (See Sec. 4.2 for definitions of metrics.) We
report the runtime per simulation step of each iteration for both methods. The error metric of the material parameters is
the Frobenius norm of the difference normalized by the Frobenius norm of the target. The error of the simulated result
is defined by the average pairwise vertex distance normalized by the size of the cloth. The numbers in each cell are
mean values with the standard deviation across 10 sample materials. Our method achieves better results and runs faster.

**The mathematical derivation and notations (Reviewers #1, #2):** We will revise them as suggested. The detailed
derivation will be provided in the supplementary document.

**Characterization of control task (Reviewer #1).** The initial control force is set to zero. The control network consists
of two FC layers, where the input (size $81 * 2 * 3$) is the position and velocity of each vertex, the hidden layer is of size
200, and the output is the control force (size $4 * 3$). The learning rate is $10^{-4}$ and the momentum is $0.5$. The reported
result is the best among 10 trials. We will provide these and other implementation details in the supplement.

**Cloth-body interaction (Reviewer #1).** As mentioned in Sec. 3.2 and 3.4, the cloth-body interaction is achieved by
continuous collision detection using a bounding volume hierarchy (BVH), and collision response using impact zone
optimization. It can be integrated into other simulations as long as the corresponding mesh BVH is used, which is often
the case.

**Comparison to regular simulators (Reviewer #1).** Our contribution to the efficiency is mostly in the backward
propagation phase, which regular simulators do not have. Our simulator is designed to be embedded in deep networks.
When gradients are needed, our simulator shows significant improvement over finite difference methods, as discussed
above. Regular simulators need to run one simulation for each input variable to compute the gradient, while our method
only needs to run once for all gradients to be computed. Therefore, the more input variables there are during learning,
the greater the performance gain that can be achieved by our method over finite difference methods.

**Relationship between Algorithm 1 and QP (Reviewer #2).** Algorithm 1 is a general flow of physical simulations.
During the collision response phase, a set of linear constraints needs to be satisfied to avoid collision. In order to
introduce minimum change to the original mesh state, we develop a QP problem to solve for the constraints. We will
provide a general tutorial of physical simulation as a QP problem in the supplement.

**Collision-rich applications (Reviewer #3).** We will provide additional examples of collision-rich motion control in
the final version.

**Discussion on frictional contacts (Reviewer #3).** Frictional contacts are modeled using frictional forces when two
objects are in close proximity and have relative motions. The description was omitted since the formulation for adding
frictional forces is standard. We are happy to add the description in the final version.

| Method | Runtime (sec/step/iter) | Density Error (%) | Non-Ln Streching Stiffness Error (%) | Ln Streching Stiffness Error (%) | Bending Stiffness Error (%) | Simulation Error (%) |
|---|---|---|---|---|---|---|
| Wang *et al*. | $2.89 \pm 0.02$ | $4.2 \pm 5.6$ | $64 \pm 34$ | $72 \pm 90$ | **$70 \pm 43$** | $4.9 \pm 3.3$ |
| Ours | **$2.03 \pm 0.06$** | **$1.8 \pm 2.0$** | **$57 \pm 29$** | **$45 \pm 41$** | $77 \pm 36$ | **$1.6 \pm 1.4$** |

Table 1: Results on the material parameter estimation task. Results with lower values have higher accuracy. 'Ln' stands for 'linear' in the table. Our method achieves higher accuracy with faster runtime in comparison to Wang *et al*., which uses finite differences for gradient computation.

[Meta-Review · NeurIPS 2019]

The reviewers unanimously recommended to accept this work --- congratulations! In your revision, please address all requests for filling in missing derivations/details (see reviewer 1's comments and others). Of course address all promises from the rebuttal phase as well. After the review process, this paper sparked some discussion on whether it really belongs at a machine learning conference, since it is quite close to the "adjoint method" for simulation and examples 4.1-4.2 are not technically involved from a machine learning perspective. If possible in the revision, any additional experiments/applications/examples that can be added that justify inclusion at a learning conference (as opposed to simulation/graphics) would help.